# Welfare Economic Analysis of Lifting Water Subsidies for Banana Farms in Jordan

**Tala H. Qtaishat** [1] , **Mohammad S. El-Habbab** [2] **and Dan P. Bumblauskas** [3],*

1   Department of Agricultural Economics and Agribusiness Management, School of Agriculture,
    The University of Jordan, Amman 11942, Jordan; t.qtaishat@ju.edu.jo
2   Financial & Agribusiness Freelance Consultant, Amman 11942, Jordan; samirhabbab@gmail.com
3   Department of Management, College of Business Administration, University of Northern Iowa, Cedar Falls,
    IA 50614-0123, USA
*   Correspondence: daniel.bumblauskas@uni.ed; Tel.: +1-96-265-355-000

**Abstract:** Jordan is one of the four driest countries in the world. Due to rapid population growth, water demand distinctly exceeds supply. The tariff to cover operations and maintenance (OM) and depreciation costs will be JD 0.066 per cubic meter (1 JD = 1.41 US$) if billing and collection efficiencies were able to reach 100 percent. The current charges of irrigation water in the Jordan Valley are JD 0.011/M$^3$. This study aims at evaluating the effect of lifting the irrigation water subsidy for bananas in the Jordan Valley on the producers' income, the consumers' expenditure, the government's budget, and Jordanian society. The total area planted with banana trees in 2017 was 1533 ha producing about 73.9 thousand tons of bananas. Previous studies have focused on evaluating the effect of input subsidies on prices and quantities, while in this study we evaluate the monetary effect on lifting these subsidies. A partial market equilibrium model, which evaluates the consumer surplus and producer surplus, was used to analyze the welfare effect of lifting the subsidies for irrigation water for the banana farms in Jordan. All the relevant players in the irrigated banana sector in Jordan were analyzed in this study including: producers, consumers, taxpayers, and society. Welfare economic instruments such as consumer surplus, producer surplus, and economic efficiency have been applied in the analysis. The government revenue will increase during the selected years by JD 3.1 million, JD 4.5 million and JD 3.8 million respectively. The dead weight loss during the three years will be JD 23.2 thousand, JD 44.3 thousand and JD 38.6 thousand respectively.

**Keywords:** welfare; water subsidy; Jordan; efficiency; banana; agricultural sustainability; Operations & Maintenance

---

## 1. Introduction

Approximately 91%, of Jordan's territory is classified as desert. Summers are generally hot and dry, while winters can be cold especially in western parts of the country. The annual rainfall varies from about 30 mm in desert areas up to 572 mm in the hilly northwest of Jordan. Almost all precipitation falls between November and May. In the Jordan Valley, winters are mild and summers very hot, with very little rainfall throughout the year.

According to the Ministry of Water and Irrigation's (MWI) National Water Strategy Policy [1] Jordan is one of the four driest countries in the world. Due to rapid population growth, water availability per capita has declined significantly, from 3600 m$^3$ per capita in 1946 to only 130 m$^3$ in 2016. Water demand distinctly exceeds supply, creating operation, maintenance, and supply chain management concerns. Almost two thirds (64%) of available water is supplied for irrigation, while the remaining 36% is broken down as municipal use which accounts for 30%, industry for 5% and tourism

for [1]. In order to overcome the water crisis, the Jordanian National Water Strategy Policy focuses on demand management and an increase in water supply through the utilization of treated wastewater, the exploitation of the non-renewable Disi aquifer [2] and a canal from the Red Sea to the Dead Sea. Table 1 shows that irrigation the water distribution efficiency and selling efficiency in 2015 were 88% and 89% respectively.

**Table 1.** Water Efficiency in Irrigation Distribution Systems.

| Item | 2014 | 2015 |
|---|---|---|
| Water Quantity Released (MCM) | 189 | 206 |
| Quantity of Water for Irrigation Order (MCM) | 164 | 181 |
| Distribution Efficiency | 87% | 88% |
| Water Quantity Sold (MCM) | 162 | 183 |
| Selling Efficiency | 86% | 89% |

Source: Jordanian Ministry of Water and Irrigation [2].

The current charges of irrigation water in the Jordan Valley are JD 0.011/M$^3$, while the total cost (in JD per M$^3$) is on the basis of quota (or actual) volume (with cross-subsidies) while assuming 100% billing and collection efficiency. However, the billing and collection efficiencies are at 82 percent [3]. The tariff to cover operations and maintenance (OM) and depreciation was JD 0.066 per cubic meter if billing and collection efficiencies reached 100%.

The World Trade Organization (WTO) asks it members to eliminate all price distortions of commodities caused by governmental interventions. One of these distortions is input subsidies. Most of the studies focuses on evaluating the effect of input subsidies on prices and quantities, while our research question to be answered is what impact the monetary 68 effect has on lifting these subsidies.

Many researchers have previously studied the distortion effect on producers, consumers, the government and the community by providing producers' subsidies to the producers. One of these studies was conducted to evaluate the technical efficiency of food processing firms by using stochastic frontier analysis and applying a true fixed effect model on production function [4]. They found that an increase of subsidies causes a slight increase in the mean of technical inefficiency. In addition, the technical efficiency of the firms without subsidies was higher than in the subsidized firms and differs, with statistical significance, with respect to time and the region of the firm.

Another study used nationally representative two-wave Integrated Household Panel Survey (IHPS) data from 2010 and 2013 [5]. In their study, the researchers employed fixed effect and correlated random effect quantile regression models were employed to estimate the conditional mean and the heterogeneous effects of subsidized fertilizer. The study found a positive effect of subsidized fertilizer on the availability of kilocalories per capita per day, the number of months of household food security, and the probability of a household being food secure over the whole year. The study also found heterogeneous effects of the program with relatively higher impact on food secure households. However, the study found no evidence of effects on the annual per capita consumption expenditures [5].

Another study used an econometric model to evaluate the impacts of subsidized hybrid seed on indicators of economic well-being among smallholder maize growers in Zambia [6]. The study found that subsidy recipients were worse off than non-subsidy recipients who planted maize hybrids and those households that did not plant maize hybrids were poor [6].

Another study related to water policy in Jordan found that water demand exceeds supply and the gap is expected to increase [7]. Moreover, they found that surface water delivered to Jordan via the Jordan River is falling in both quantity and quality measures, while irrigation water in the Jordan Valley is becoming increasingly saline. In addition, some of the water is used in agriculture for relatively low value-added production output. At the same time, it is known that bananas are one of the largest fruit

trees water consuming in Jordan. Accordingly, it is necessary to regulate water consumption for this crop. One of the policies used to tackle this subject is eliminating the irrigation water subsidy.

Riley defined "producer subsidies," provides a listing of different types of producers' subsidies, and then discusses the effect of the consumer subsidy using producer and consumer surplus tools [8]. The MWI in Jordan emphasized the importance of water demand management since the observed water consumption was at least partly a result of the limited water supply [9] bias. In 2011, the irrigation water provided by the MWI was less than the crop requirements in the Jordan Valley. The average gross value added per $m^3$, i.e., for the last $m^3$ that goes to a sector of water consumption, in fruit tree production was 0.23 and in vegetables was 1.23. However, availability does not necessarily reflect the actual water demand [10].

Water and sewer services revenues only cover a portion of the overall O&M costs in Jordan, in part due to the increase of electricity tariff costs in 2011 by 14%. Subsidies to the water sector amount to more than 0.4% of the GDP [11]. In 2015, Jordan consumed about 1009 million Cubic Meters of water, out of which the agriculture sector used 53% of Jordan's water while the domestic sector used 42% and the remaining 5% was used by the industrial sector [2]. Although Jordan suffers from water shortages, it still subsidized the irrigation water which encouraged over use of water in 2008 [12].

Other researchers have estimated the elasticity of demand of irrigation water in the Jordan Valley at −0.215 using a linear programming model [13]. A study that estimated the cost of irrigation water in Jordan found that the O&M costs were JD 0.063/$M^3$, and the collection ratio was 60%. The study recommended that the Jordan Valley Authority (JVA), the institution managing the valley's water supply, could increase billing and collection efficiency and ensure that all farmers who use JVA water pay for the service. In addition, farmers can expect to see improved energy efficiency and cost efficiency [3].

Many researchers have used the partial equilibrium model (PEM) to evaluate policy impacts of this sector. Reference [14] used the model in agricultural commodity market outlook models, many researchers have used the partial equilibrium model (PEM) to evaluate policy impacts of this sector. Reference [14] used the model in agricultural commodity market outlook models, such as the IMPACT model of International Food Policy Research Institute (IFPRI) and the Rice outlook model of International Rice Research Institute (IRRI). Moreover, Food and Agricultural Organization of the United Nations (FAO) used this model for generating short—Run and medium—Term outlook for major food commodities [15]. In 2007 researchers, used a partial equilibrium trade model to evaluate the Effects of U.S. Distortions in the Ethanol Market. They used an export supply function for Brazil and an ethanol import demand function for the United States of America (USA). They found that the USA and Brazil would both realize trade [16].

## 2. Research Motivation

The O&M expenses for the JVA are much higher than the value of charges paid by the farmers; this budgetary deficit was about JD 6 million in the year 2012 [3]. The main causes of the deficit were the aging infrastructure and along with low billing and collection efficiencies. These factors have placed a heavy financial burden on the Jordanian government's budget. This study aims at evaluating the effect of lifting irrigation water subsidies for bananas in the Jordan Valley and the associated impact on the producers' income, consumers' expenditure, the government budget and Jordanian society.

*Banana Production in Jordan*

The total area of planted bananas in Jordan in 2017 was 15,330 dunums (1 dunum = 0.1 ha); while the total production in the same year was about 73.9 thousand tons (Table 2).

Table 2 shows the total area planted with banana trees in 2017 was 15330 dunums producing about 73.9 thousand tons. Figure 1 shows that about 82% of banana production in Jordan was in South Shounah, followed by North Shounah (10%), and finally Ghor Al-Safi (8%).

**Table 2.** Area and Production of banana in the Ghor area in Jordan in 2017.

| District | Area (Dunum) | Production (Ton) |
| --- | --- | --- |
| North Ghor | 2600 | 7700 |
| South Shounah | 12,100 | 60,500 |
| South Gors (Ghor Al-Safi) | 630 | 5650 |
| Total | 15,330 | 73,850 |

Source: [17].

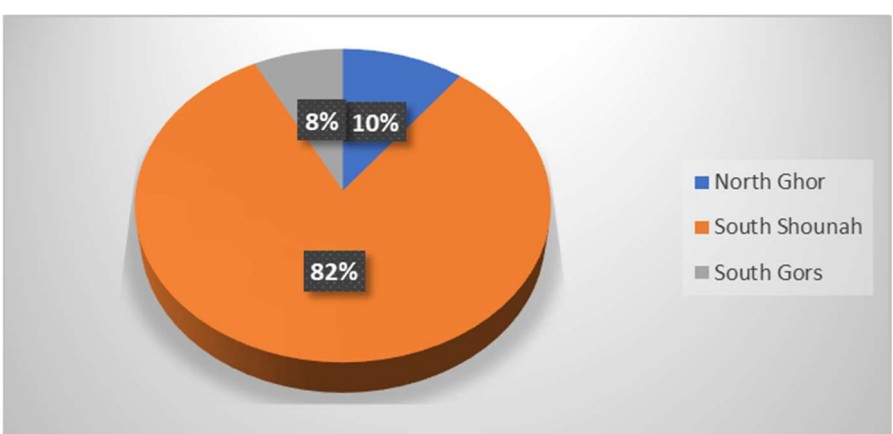

**Figure 1.** Banana Production in Jordan in 2017.

## 3. Methodology

### 3.1. The Conceptual Model

The partial market equilibrium modeling method was used to analyze the welfare effect of lifting the subsidies for irrigation water for the banana farms in Jordan. All the relevant players in the irrigated banana sector in Jordan were analyzed in this study, including: producers, consumers, taxpayers, and society. Welfare economic instruments including consumer surplus, producer surplus and economic efficiency were applied in the analysis.

One rationale for using partial equilibrium modeling (PEM) is that this modeling approach requires minimal data. The only required data include are the trade flows, the trade policy such as subsidies and tariffs, and a couple of behavioral parameters such as elasticities of supply and demand. However, the primary and most important rational for using this modeling technique is that it permits an analysis at a fairly disaggregated (or detailed) level which resolves the aggregation bias. The users can employ the basic concepts introduced in PEM to create more complex PEMs for policy analysis.

Most scholars have historically used PEM in analyzing trade policies, but in this study it is being utilized as a policy model to evaluate the subsidy effect elimination on the entire Jordanian economy. Accordingly, the main hypotheses of this study can be stated as follows:

**Hypothesis 1.** *The elimination of the irrigation water input subsidy will release the expenditure burden on the government budget*.

**Hypothesis 2.** *The elimination of the irrigation water input subsidy will decrease banana product water consumption, due to a decrease in its production.*

The required data for the analysis included the following: elasticity of supply and elasticity of demand for bananas, domestic wholesale prices, border prices, import and export quantities of bananas, and the official exchange rate. Most of this data were collected from Reference [18]. The analysis evaluated the following indicators:

1.  The effect of the producer subsidy on consumer and producer surplus and the gains and losses realized by the government along with the gains and losses realized by society, i.e., dead weight loss.
2.  Partial equilibrium indicators which are used here to assess the impact of a price intervention or of policies that shift the supply and/or demand curves. These include:

   ■　Welfare Effects: These effects are evaluated by measuring the change in consumer surplus ($\Delta$CS) and the impacts on producer welfare relative to the change in producer surplus ($\Delta$PS). It covers both, consumer groups and producer groups, where they have different price elasticity of demand and supply.

   ■　Government Budget Effect ($\Delta$B**):** Although there are two important sources of government revenues, which are import tariffs and export taxes, in Jordan there are no export taxes. In this study the effect of lifting water subsidies, which will decrease the burden on the government budget, was evaluated.

   ■　Efficient Effects: The net social gain or loss (NSG, NSL) to a country was measured in this study as the total of the changes in consumer surplus, producer surplus, government budget effect, and rent effect. As we will see below, an NSL can typically be decomposed between a NSL in production (NSLP) and a SNL in consumption (NSLC).

### 3.2. Analytical Framework of the Partial Market Equilibrium Model

Figure 2 provides a graphical representation of the model developed in Section 3.1.

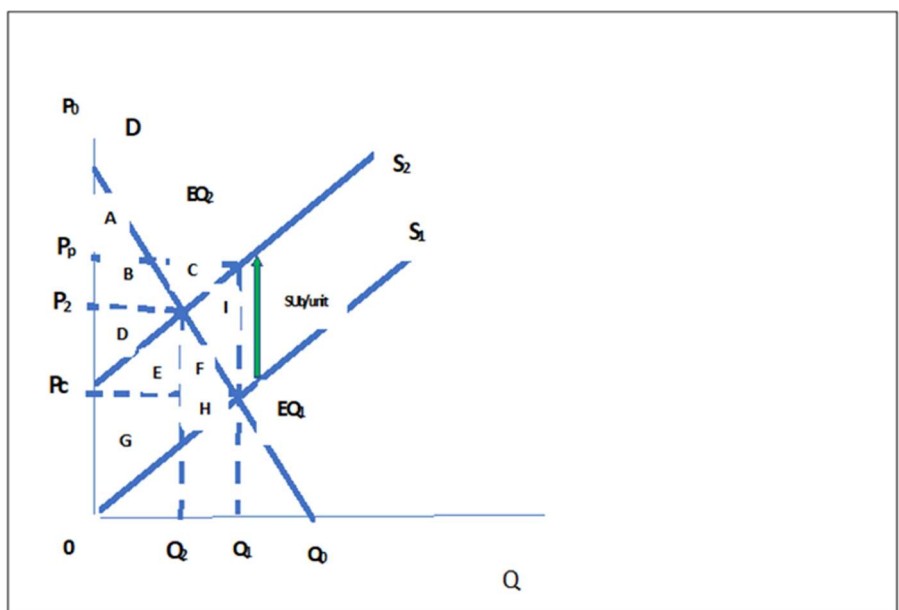

**Figure 2.** Graphical Representation of Conceptual Model. Where: $P_1$: Wholesale price with subsidy = Pc. $P_2$: Wholesale price without subsidy. Pp: Producer price without Subsidy. DWL: Dead weight loss. CS: Consumer surplus. PS: Producer surplus. Subsidy = Water subsidy converted to JD/ton of banana production = Pp – Pc.

$$\Delta P = (P_2 - P_1)/P_1\% \tag{1}$$

$$\Delta\, CS = (P_2 - Pp) * (Q_2 - Q_1)/2 \tag{2}$$

$$\Delta\, PS = (P_2 - Pc) * (Q_2 - Q_1)/2 \tag{3}$$

$$\Delta\, PS = (P_2 - Pc) * (Q_2 - Q_1)/2 \tag{4}$$

$$\text{Government budget: Total Subsidy} = Q_1 * (Pp - Pc) \tag{5}$$

$$\text{Dead Weight Loss DWL} = 0.5 * (Pp - Pc) * Q_1 \tag{6}$$

The Nominal Protection Coefficient (NPC) is calculated by dividing the domestic wholesale price of the bananas to its border price using the official exchange rate:

$$NPC = pd/pb \tag{7}$$

where:

Pd: domestic price, and
Pb: border price

Alternatively, this can be written as the nominal rate of protection (NRP).

For the purpose of quantitative analysis, it is important to measure the efficiency, welfare, government revenue, and balance of trade effects as follows:

The net social loss in production (NSLP), which can be measured as follows:

$$NSLP \; = \; -0.5 * \left( \frac{Q_1 - Qb}{p1 - pb} * \frac{Pb}{Qb} \right) \; * \; \left( \frac{Qb}{Pb} (P_1 - Pb) * (Pb - P) \right) \; = \; -0.5 * \varepsilon s \; * \; \left( \frac{P - Pb}{Pb} \right) * Pb * Qb \tag{8}$$

where:

$$Es = [(Q - Qb)/(P - Pb)] \, (Pb/Qb) \tag{9}$$

Efficiency effects:

$$NSLP = -0.5 * \varepsilon s * S * Pb * Qb \tag{10}$$

Net Social Loss in Consumption (NSLC) =

$$NSLC = 0.5 * \varepsilon s * Qs2 * Pb * Pb \tag{11}$$

Net Social Loss (NSL) =

$$NSL = NSLP + NSLC \tag{12}$$

The degree to which prices influence consumed and produced quantities is as follows:

- The percentage difference between world (border) price and producer price was calculated using the following formula:
$$[\{1 - NPC \, (Pp)\}/NPC \, (Pp)] * 100 \tag{13}$$

- The percentage difference between border price and consumer price was calculated using the following formula:
$$[\{1 - NPC \, (Pc)\}/NPC \, (Pc)] * 100 \tag{14}$$

*3.3. Data Collection*

The research was built on secondary data, mainly FAOSTAT (2018). Table 3 shows the main data that was required to run the conceptual model:

**Table 3.** Basic information for banana sector in Jordan.

| Item | 2014 | 2015 | 2016 |
|---|---|---|---|
| Es (Elasticity of supply) | 1.58 | 1.58 | 1.58 |
| Ed (Elasticity of demand) | −0.215 | −0.215 | −0.215 |
| Border Price (US$/Ton) | 887 | 795 | 800 |
| Exchange Rate | 0.7008 | 0.7008 | 0.7008 |
| Border Price (JD/Ton) | 622 | 557 | 561 |
| Local Production (Ton) | 37,489 | 46,835 | 40,857 |
| Imports (Ton) | 47,760 | 50,353 | 39,253 |
| Exports (Ton) | 124 | 132 | 755 |
| Quantity Demanded (apparent Consumption) Ton | 85,125 | 97,056 | 79,355 |
| Quantity Supplied (Ton) | 85,249 | 97,188 | 80,110 |
| Water Subsidy JD/ton | 44.72 | 44.72 | 44.72 |

The elasticity of supply was calculated from the marginal cost function derived from [19]. The estimated long-run cost, average cost and marginal cost functions were as follows:

$$\text{LRTC} = 0.23Q - 0.00004Q^2 + 0.0000000023Q^3 \tag{15}$$

$$\text{LRAC} = 0.23 - 0.00004Q + 0.0000000023Q^2 \tag{16}$$

$$\text{MC} = 0.23 - 0.00008Q + 0.0000000069Q^2 \tag{17}$$

where:

LRTC: Is long term total cost function
LRAC: Is long term average cost function
MC: Is marginal cost function
Q: Quantity produced

Since the banana market in Jordan is nearly considered a perfect competition market, the supply function could be derived by implementing prices in the marginal cost function, and the average elasticity of supply could be derived by the following relationship:

$$\text{Es} = \Delta Q * \hat{P}/\hat{Q} \tag{18}$$

where $\Delta Q$ is the value of the supply function slope, $\hat{P}$ and $\hat{Q}$ are the average price and average quantity respectively. The elasticity of supply (Es) was estimated at 1.58. The elasticity of demand for the bananas was estimated by the team using Reference [18] prices and quantity data at −0.187, i.e., inelastic demand.

Border prices in USD ($) for the year 2016 were collected from Reference [18]. These prices were then converted by the official exchange rate which is fixed at 0.7008 JD/1 $US. Local production, imports and exports were also collected from Reference [18]. Table 4 shows the balance sheet for bananas in Jordan during 2010–2016 [18].

**Table 4.** The Balance Sheet for Banana in Jordan during 2010–2016.

| Year | Production (Tons) | Import (Tons) | Export (Tons) | Apparent Consumption (Tons) | Per/Capita Consumption Kg/Person | Population |
|---|---|---|---|---|---|---|
| 2010 | 43,753 | 40,206 | 1296 | 82,663 | 12.5 | 6,594,000 |
| 2011 | 48,304 | 64,167 | 441 | 112,030 | 16.4 | 6,846,000 |
| 2012 | 38,852 | 51,423 | 477 | 89,798 | 12.5 | 7,210,000 |
| 2013 | 42,008 | 43,462 | | 85,470 | 11.0 | 7,771,000 |
| 2014 | 37,489 | 47,760 | 124 | 85,125 | 10.1 | 8,459,000 |
| 2015 | 46,835 | 50,353 | 132 | 97,056 | 10.6 | 9,182,000 |
| 2016 | 40,857 | 39,253 | 755 | 79,355 | 8.1 | 9,798,000 |

Source: [18].

The apparent consumption is calculated by adding imports to local production then deducting exports, while per capita consumption was calculated by dividing the apparent consumption by the population. The water subsidy was evaluated according to the following process steps:

1. Looking up the value of the subsidy for water from a report titled "The Cost of Irrigation Water in the Jordan Valley." It turns out to be JD 0.0559/cubic meter of irrigation Water [3].
2. The irrigation water used for banana and yield per dunum in Jordan were published by the Agricultural Credit Corporation. The irrigation water applied to bananas was 2000 M$^3$/dunum and the yield was 2.5 tons/dunum.
3. From this information the value of subsidy for one ton of bananas produced in Jordan was estimated at JD 44.72/ton.

## 4. Results and Discussions

Table 5 shows the method of calculation of border, wholesale and retail price of bananas, while taking into consideration marketing costs.

**Table 5.** Banana Marketing Costs Calculations (per ton).

| | 2014 | 2015 | 2016 |
|---|---|---|---|
| Border Price (JD/Ton) | 457 | 584 | 607 |
| Import tax (4%) | 18 | 23 | 24 |
| Whole Sale Price (JD/Ton) | 439 | 560 | 583 |
| Market fees (4%) | 18 | 22 | 23 |
| Commission (5%) | 22 | 28 | 29 |
| Transportation | 10 | 10 | 10 |
| Retail (Consumer) Price (JD/Ton) | 488 | 621 | 645 |

The results from conducting the partial equilibrium model analysis for bananas are displayed in Table 6. The quantity supplied with subsidy was 85.2 thousand tons in 2014, about 97.2 thousand tons in 2015, and about 80.1 thousand tons in 2016. After eliminating the water subsidy, the quantity supplied became (for the same time periods) 40.5 thousand, 50.8 thousand and 44.2 thousand tons. On the other hand, the quantity demanded with a subsidy was 85.125 thousand, 97.065 thousand and 79.355 thousand tons in 2014, 2015, and 2016 respectively, and changed to 86 thousand, 98 thousand and 80.1 thousand during the same years respectively after lifting the subsidy.

The efficiency loss in production is JD 34.5 thousand, JD 58.5 thousand and 50.3 Thousand during the years 2014–2106 respectively, while Efficiency gain in consumption during the same three years was JD 11.4 thousand JD14.3 thousand and JD11.7 Thousand. The government revenue will increase during the selected years by JD 3.1 million, JD 4.5 million and JD 3.8 million respectively.

The dead weight loss during the three years will be JD 23.2 thousand, JD 44.3 thousand and JD 38.6 thousand respectively.

**Table 6.** Results of the Partial Market Equilibrium Model for Bananas in Jordan.

| Item | Year | | |
|---|---|---|---|
| **ASSUMPTIONS:** | **2014** | **2015** | **2016** |
| Elasticity of Supply | 1.58 | 1.58 | 1.58 |
| Elasticity of Demand | −0.187 | −0.187 | −0.187 |
| Production (MT) | 37489 | 46835 | 40857 |
| Exports (MT) | 124 | 132 | 755 |
| Imports (MT) | 47,760 | 50,353 | 39,253 |
| Net Trade (MT) | 47,636 | 50,221 | 38,498 |
| Apparent Consumption (MT) | 85,125 | 97,056 | 79,355 |
| World Prices Pw (US$/MT) | 652 | 833 | 866 |
| Exchange Rate (JD/$US) | 0.708 | 0.708 | 0.708 |
| Border Price (JD/MT) | 462 | 590 | 613 |
| Producer Price (PPd), (JD/MT) | 439 | 560 | 583 |
| Consumer Price (PCd), (JD/MT) | 488 | 621 | 645 |
| NPCp | 0.95 | 0.95 | 0.95 |
| NPCc | 1.06 | 1.05 | 1.05 |
| **SHIFT TO ELIMINATING SUBSIDY** | | | |
| Movement from PPd to Pw (%) | 5.152 | 5.315 | 5.168 |
| Movement from PCd to Pw (%) | −5.407 | −5.030 | −4.941 |
| Change in production (MT) | 3051 | 3933 | 3336 |
| Change in Consumption (MT) | 861 | 913 | 733 |
| **AFTER ELIMINATING WATER SUBSIDY** | | | |
| Supply (MT) | 40,540 | 50,768 | 44,193 |
| Demand (MT) | 85,986 | 97,969 | 80,088 |
| Net Trade (MT) | 45,445 | 47,201 | 35,895 |
| Efficiency Loss in Production (JD) | (34,506) | (58,532) | (50,253) |
| Efficiency Gain in Consumption (JD) | 11,354 | 14,258 | 11,685 |
| Total Deadweight Loss (JD) | (23,153) | (44,274) | (38,568) |
| Consumer Gain/Loss (JD) | 2,279,999 | 3,074,415 | 2,564,259 |
| Producer Gain/Loss (JD) | (882,357) | (1,452,529) | (1,281,193) |
| Government Revenue (JD) | 3,116,496 | 4,454,154 | 3,783,513 |

## 5. Conclusions and Future Work

This article used a partial equilibrium modelling to evaluate lifting irrigation water subsidies for bananas in the Jordan Valley. The results showed that the producers will sustain a decrease in revenues due to the decrease in production (Hypothesis 2) and that the consumers will gain from eliminating subsidy on bananas. On the other hand, the government will gain from the suggested policy of lifting the subsidies on water for bananas at the amounts of JD 3.1 million, JD 4.5 million, and JD 3.8 million in 2014, 2015, and 2016 respectively (Hypothesis 1).

The calculated efficiency loss of the banana production sector was relatively high related to this sector. The implementation of this policy (i.e., lifting the subsidy on water prices) will save water, since the production of banana will decrease.

The study suggests eliminating the Jordanian water subsidy to cover at least the OM costs, so as to lift the burden on the government budget, and to convince farmers to use irrigation water more efficiently. By applying good maintenance practices for aging infrastructure and implementing appropriate procedures for increasing the billing and collection efficiencies, gains can be achieved across the country.

**Author Contributions:** T.H.Q. and M.S.E.-H. designed the model and collected the data. D.P.B. helped in reviewing some articles and literature review. T.H.Q. carried out the economic analysis. All three authors drafted and revised the manuscript.

**Funding:** This research received no external funding.

**Acknowledgments:** The authors are grateful to both the Ministry of Water and Irrigation and the Ministry of Agriculture in Jordan for providing us with the data used in the article.

**Conflicts of Interest:** The authors declare no conflict of interest.

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
