# Peer review of "Welfare Economic Analysis of Lifting Water Subsidies for Banana Farms in Jordan"

_sustainability, doi:10.3390/su11185118_

Round 1

Reviewer 1 Report

This is a quite good article. However, the conclusion is weak and lacks detailed elements and justifications. 

Please revise this 'Selling Efficiency % 86 89 ' in Table 1

Also, the article doesn't integrate climate projections so I am not sure about the suitability of its publication in this journal. 

Reviewer 2 Report

Limitation 

In discussing the results, it is instructive to compare with other studies from other countries in the region to be able to make an informed comparison on the economic impact on both the farmers and the government.

The methodology has not shown the size of the data used for the study. This is useful for readers to be able to put in scope and perspective the results and conclusion.

I would suggest better exploring the lesson and implications on other sector of the value chain as it is currently limited to farmers and the government. Also, what is the policy element of the study, which is currently not supported by the data.

Is there any rationale as to why the Partial Market Equilibrium Model for Bananas in Jordan is only calculated for 2014, 2015 and 2016 in table 6?

Can you indicate from the analysis how the Government will gain from the suggested policy for lifting the subsidies on water for bananas? It only appeared in the conclusion
